# Polyphenol Composition and Antioxidant Potential of Instant Gruels Enriched with *Lycium barbarum* L. Fruit

**DOI:** 10.3390/molecules25194538

**Published:** 2020-10-03

**Authors:** Marta Olech, Kamila Kasprzak, Agnieszka Wójtowicz, Tomasz Oniszczuk, Renata Nowak, Monika Waksmundzka-Hajnos, Maciej Combrzyński, Marek Gancarz, Iwona Kowalska, Anna Krajewska, Anna Oniszczuk

**Affiliations:** 1Department of Pharmaceutical Botany, Medical University of Lublin, 20-093 Lublin, Poland; renata.nowak@umlub.pl; 2Department of Inorganic Chemistry, Medical University of Lublin, 20-093 Lublin, Poland; monika.waksmundzka-hajnos@umlub.pl; 3Department of Thermal Technology and Food Process Engineering, University of Life Sciences in Lublin, 20-612 Lublin, Poland; maciej.combrzynski@up.lublin.pl; 4Institute of Agrophysics, Polish Academy of Sciences, 20-290 Lublin, Poland; m.gancarz@ipan.lublin.pl; 5Department of Biochemistry and Crop Quality, Institute of Soil Science and Plant Cultivation, State Research Institute, 24-100 Puławy, Poland; ikowalska@iung.pulawy.pl; 6Department of Integrated Paediatric Dentistry, Medical University of Lublin, 20-094 Lublin, Poland; anna.krajewska1@umlub.pl

**Keywords:** dietary polyphenols, liquid chromatography, functional food, instant gruels, goji fruit, antioxidant activity, extrusion-cooking, processing parameters

## Abstract

Goji fruit (*Lycium barbarum* L.) has been identified as a polyphenolic compound plant source of noted richness. It also contains polysaccharides, carotenoids, vitamins and minerals, fatty and organic acids. The purpose of the presented research was to produce innovative instant corn gruels with various dry goji berry contents (1, 3 and 5%), to determine the level of included polyphenolic compounds (including individual free phenolic acids) and to assess the antioxidant properties of these functional-food products. A further objective was to identify the optimum value of one of the most important production parameter, the rotational speed of the extruder’s screw during gruel processing. The undertaken chromatographic analysis (LC-ESI-MS/MS) showed a wide variety of available phenolic acids. In the samples with 5% addition of fruit, eight phenolic acids were detected, whereas in the corn gruel without additives, only five were noted. The antioxidant activity, the content of free phenolic acids and the sum of polyphenols increased with increase of the functional additive. For all goji content, screw speeds of 100 and 120 rpm rather than 80 rpm resulted in higher polyphenol amounts and greater Trolox equivalent antioxidant capacity, as well as higher ability to scavenge DPPH.

## 1. Introduction

Worldwide studies have shown the huge impact of food quality on the human body [1]. For this reason, functional food has become a part of the daily diet of the average consumer. The concept of functional food appeared in the 1980s in Japan in an attempt to improve the overall health of its citizens so as to control rising healthcare costs. Due to the fact that different countries have adopted this idea unevenly, there is no unequivocal definition of functional food [2]. Still, the underlying principle is that apart from basic balanced nutrition, functional foods must fulfil physiological and metabolic functions aimed at ensuring physical and mental well-being. Therefore, they represent a promising prospect for preventing obesity, diabetes, cardiovascular diseases and cancer.

Designed functional foods are products from which harmful substances have been removed, or that have been enriched with one or more modified ingredients that will give them some desired nutritional value. It should be understood that products of this type must resemble ordinary conventional food and cannot be tablets, drops or any other form of a drug. Furthermore, they have to exert their beneficial effects in the amount available for general consumption as a supplement to the daily diet [3].

Extracts and plants rich in polyphenols are promising ingredients for enriching functional foods. Many health beneficial effects (e.g., antioxidant, antibacterial, chemopreventive, antiviral, and anti-inflammatory) have been claimed for plant-derived food products rich in phenolic acids (both benzoic and cinnamic acid derivatives) [4,5]. Hence, the prevention and treatment of many diseases is built upon the action of polyphenols [6,7]. However, the amount of these (including phenolic acids) may be affected by the processing of raw material. Therefore, monitoring of phenolic acids is commonly performed during investigation and quality control of new functional foods or plant-derived food components.

Goji fruits (*Lycium barbarum* L.) were identified as a rich source of polyphenolic compounds, therefore the beneficial effects of their consumption have been the subject of much research. Goji berries were reported to contain pharmacologically active phenolic acids (PAs), including benzoic and cinnamic acid derivatives, e.g., hydroxybenzoic acids, caffeic and *p*-coumaric acid to a significant fraction (32.7 mg/g of their dry weight) [8]. The berries were also found to contain polysaccharides and carotenoids, mainly zeaxanthin. Other important ingredients are thiamine, riboflavin and ascorbic acid. In addition, the berries contain many available minerals such as potassium, sodium, phosphorus, magnesium, calcium and iron. What is more, the fruits contain fatty acids (hexadecanoic and linoleic) and organic acids (citric, fumaric, apple and shikimic) [9,10,11].

Due to the described composition, goji berries are a natural source of antioxidants [12]. Their consumption can modulate the immune system, inhibit inflammation and positively affect the metabolism. With diabetic patients, the inclusion of goji fruits in their diet has had demonstrated hypoglycaemic and lipid-lowering effects. Moreover, goji berries can be used in the prevention of neurodegenerative diseases, mainly due to the neuroprotective effect of the contained polysaccharide complex [13,14,15].

The aim of the research was to produce innovative corn gruels with varied percentages of dry goji fruit and to determine the optimum extruder screw rotational speed for obtaining product of the highest quality. For this purpose, the antioxidant properties and polyphenolic content were investigated in all newly created extruded gruels. Moreover, liquid chromatography/electrospray ionization triple quadrupole mass spectrometry (LC-ESI-MS/MS) was used to determine phenolic acid content in the final food product.

## 2. Results and Discussion

### 2.1. Influence of Goji Fruits Addition on Polyphenols Content and Antioxidant Properties of Extruded Gruels

Phenolic compounds are a widespread group of secondary metabolites, and many desirable biological effects depend on their presence in consumed plants. In the first stage of the study, we examined the total content of polyphenols. The results showed that the polyphenol content increased significantly (*r* = 0.9704, *p* = 0.001) along with the increased functional additive content (Table 1). However, increasing the screw speed during instant gruel plus goji fruits processing had no significant effect on the content of TPC. The highest total content of polyphenols (as per gallic acid equivalents; GAE) was reported in gruels with 5% addition of the goji fruits, while the lowest was seen in a sample without such functional additive.

In the next stage of the research, we determined the DPPH free radical scavenging potential of the tested samples using two methods. Firstly, measurement of antiradical activity was carried out using a UV-VIS spectrophotometer. The obtained results showed that the antioxidant activity of extruded gruels increased along with the amount of added goji fruit. The maximum free radical scavenging ability by all extracts was obtained after 15 min (Table 2).

The next stage of the research involved TLC-DPPH testing of extracts. This experiment was aimed at confirming the scavenging properties of the gruels towards DPPH. Similar results were obtained in this method as in the previous – the sample antiradical properties significantly increased along with the increase of the goji content (*r* = 0.9594, *p* = 0.001). The change of screw speed applied during the instant gruel extrusion-cooking (indicated by different letters in Table 1) was shown to slightly affect the intensity of antioxidant activity of the tested instant gruels – especially during the first ten minutes. By far, the highest DPPH free radical scavenging activity was seen in the products supplemented with the addition of 5% fruits (Table 1).

The obtained results showed that the antioxidant activity expressed as Trolox equivalent antioxidant capacity (TEAC) was positively correlated with the amount of added goji fruit (*r* = 0.9189, *p* = 0.003), and with the total content of polyphenols (*r* = 0.8697, *p* = 0.01). Increasing screw speed had only a slight effect on TEAC, especially with high content of additive. With regard to TEAC, the sample that presented the highest TEAC value was gruel with 5% of fruit addition (Table 1; TLC chromatogram – Figure 1).

Dried and freeze-dried goji fruits were previously reported to contain ferric reducing antioxidant agents with FRAP values 0.15–0.62 and 23.09 mmol Fe^2+^/kg, respectively [16,17]. High antioxidant capacity of *L. barbarum* berries was also demonstrated by the cupric ion reducing antioxidant capacity (CUPRAC) test (26.91–35.41 mg TE/g), oxygen radical absorbance capacity (ORAC) assay (180–260 μmol vitamin C equivalents/ g) and electrochemical methods [18,19].

Previous experiments have shown that aglycones have a higher antioxidant activity than their glycosidic forms or those connected by different types of bonds [20,21]. Moreover, research has indicated that the antioxidant activity of polyphenols is dependent on the number of hydroxyl groups in the molecule and can be enhanced by spherical effects, as well as by synergistic and antagonistic interactions of the compounds present in the matrix and in the extracts [22,23]. Thus, an increase in the content of polyphenolic compounds does not always entail an increase in antioxidant properties. In addition, it has been noted that the antioxidant activity of extrudates is dependent not only on the level of bioactive compounds, but also on the their composition. Korus et al. [24] observed a lower antioxidant activity for dark-red beans compared to black-brown- and cream-colored beans, even though dark-red beans extrudates exhibited higher total phenolic content compared to black brown and cream coloured beans extrudates. In the case of our innovative gruels enriched with goji fruit, the product antioxidant activity was positively correlated with the total content of polyphenols and with the amount of added fruit.

### 2.2. Influence of the Screw Speed on Polyphenols Content and Antioxidant Properties of Extruded Gruels

In the opening phase of the study, the authors compared the total content of polyphenols in instant gruels with different content of goji (0, 1, 3 and 5%) and their ability to scavenge free radicals. Polyphenols are secondary plant metabolites with a wide range of pro-health properties and many favorable biological effects are achieved owing to their presence in functional food products.

The distribution and concentration of polyphenols can differ in various plant organs [25]. In addition, polyphenol concentration can vary due to food processing or drying because of the early degradation of phenolic compounds that is favored by high temperature and prolonged exposure to heat. This brings about the release of phenolic substances through breakage of ether, ester or acetal covalent bonds [22]. Mutari et al. [20] suggest that high temperature treatment can improve the solubility of phenolic compounds leading to the breakdown of cellular structures and improving the release of phenolic compounds (including phenolic acids) previously bound to the macromolecules of the cell wall. In addition, processing can accelerate the release of phenolic compounds due to the breaking of cellular constituents. Moreover, compounds that display health properties and have a beneficial effect on the human body can be formed.

Extrusion-cooking is a process of treating plant materials (especially starchy) under specific thermal, moisture and (high) pressure conditions. The intensity of processing, especially thermal and mechanical shearing, results in a deep transformation of individual components. The treated material is already cooked inside the extruder so starch-based foods made with via extrusion-cooking become instant or ready-to-eat. Therefore, the following stage of the research focused on the influence of one extrusion-cooking condition, the screw rotational speed, on the total content of polyphenols and the antioxidant properties of analyzed extruded gruels.

The obtained findings demonstrated that for all the goji contents used screw speeds of 100 and 120 rpm resulted in a higher content of polyphenols and Trolox equivalent antioxidant capacity, as well as higher ability to scavenge DPPH than did the speed of 80 rpm (Table 1 and Table 2). According to Alonso et al. [26], the main factors stimulating the transformation of input material during the extrusion-cooking process are high temperatures and mechanical aspects related to shear forces that rise along with the increase of screw rotational speed. It is possible that the extrusion conditions (e.g., shearing forces and temperature) at 80 rpm are too mild and do not lead to the release of some phenolic compounds from the processed blends.

In some instances, antioxidant activity is observed to increase in extruded products with an increased temperature of processing. An increase in ORAC values with enhanced extrusion temperatures was observed [27]. This effect is explained due to the presence of products formed during Maillard reactions. Yilmaz and Toledo [27] showed that Maillard reaction products obtained from heated histidine and glucose have peroxyl radical scavenging activity and that this effect relates strongly with ORAC assay. Hence, it is always advisable to assess antioxidant activity by at least two procedures (e.g., Trolox equivalent antioxidant capacity and ability to scavenge DPPH) [28].

Increasingly popular in food technology for the processing of various types of materials, the extrusion-cooking process produces diversified properties within the final product. The action of temperature, pressure and shear forces on moist raw material induces profound changes in the processed matter in a very short time. Among these are enhanced digestibility of nutrients, inactivation of anti-nutritive factors and modified sensory characteristics. The intensity of these changes depends both on the properties of raw materials and on the settings of the extrusion-cooking procedure (i.e., the temperature or rotational speed of the extruder’s screw). A high degree of mixing and homogenization leads to a decrease in diffusion barriers and the breaking down of chemical bonds that result in a heightened reactivity of the components [29,30]. Therefore, the proper selection of manufacturing parameters is paramount.

### 2.3. Free Phenolic Acid Content (LC-ESI-MS/MS)

Phenolic acids are important polyphenol group of *L. barbarum* fruit. Thus, in the next stage of the research, the authors performed a quantitative analysis of free phenolic acids in instant gruels produced at 100 rpm. The LC-ESI-MS/MS method was optimized to contain PA previously reported to be found in large quantities in goji berries.

LC-ESI-MS/MS analysis demonstrated eight phenolic acids present in the 5% goji fruit samples (Table 3, Appendix A. They are derived from benzoic acid and include: protocatechuic acid, 4-OH-benzoic acid, gentisic acid, salicylic acid, and cinnamic acid derivatives: *trans*-caffeic, *p*-coumaric, ferulic, and isoferulic acid. This was the only sample where *trans*-caffeic and gentisic acids were determined. In the instant gruel without any functional additive, only five free phenolic acids were detected: protocatechuic acid, 4-OH-benzoic acid, ferulic acid, isoferulic acid and salicylic acid. The content of individual acids and their sum increased as more goji fruit was added. We observed that gruels enriched with this functional additive became an important source of free phenolic acids, especially ferulic and isoferulic acid. Their content in goji fruit enriched samples was very high and was enhanced significantly as more of this fruit was added (*r* = 0.9361, *p* = 0.001 for ferulic, and *r* = 0.7055, *p* = 0.01 for isoferulic acid).

The biological properties of goji berries have been ascribed to their high content of nutrients and phenolics. Comprehensive studies aimed at unambiguously identifying the phenolic components in goji berries are lacking [31]. Inbaraj et al. [32] have developed a high-performance liquid chromatography–diode array detection–mass spectrometry method with electrospray ionization mode (HPLC–DAD–ESI-MS) for determining polyphenols in fruits of *Lycium barbarum* L. Here, a total of 52 compounds were separated. Among these, 15 phenolic acids and flavonoids were positively identified (based on both absorption and mass spectra) and quantified. Within the positively identified compounds, phenolic acids (*p*-coumaric, caffeic, chlorogenic and vanillic) were present in large mass fractions. For this reason, the mentioned group of compounds has become the subject of our interest in goji fruit enriched instant gruels.

Apart from health benefits, phenolic acids in food also act as free radical terminators, chelators of metal catalysts, and singlet oxygen quenchers abetting lipid oxidation to improve shelf life and consumer acceptance of extruded products. It was found by Viscidi et al. [33] that the addition of ferulic acid and benzoin at levels of 1.0 g/kg or higher generally resulted in delayed onset of oxidation in oat-based extrudates.

Our results showed that the antioxidant activity of the examined food was positively correlated with the content of free phenolic acids (*r* = 0.8697, *p* = 0.01 for TEAC, and *r* = 0.9437, *p* = 0.001 for DPPH scavenging ability). The content of free phenolic acids in the examined extracts was ascertained by means of calibration curves designed for each model. The limit of detection (LOD) and the limit of quantification (LOQ) were determined at the signal to noise ratio (S/N) of 3 and 10, respectively. All the tested compounds showed positive linearity. The values of LOD and LOQ, as well as the linear range for all analyzed compounds are shown in Table 4.

These results are aligned with those of Madhujith et al. [21]. They reported that the level of free and non-bonded and conjugated phenolic acids significantly contributed to the antioxidant properties of the tested plant samples. Our research demonstrated, therefore, that innovative gruels with the addition of dry goji fruits can become a source of the antioxidants that are so important for the human body.

## 3. Materials and Methods

### 3.1. Chemicals

Acetonitrile and formic acid requested for chromatographic analysis, ethanol for extraction and Folin-Ciocalteu reagent were purchased from J.T. Baker (Phillipsburg, PA, USA). The applied standards, ABTS (2,2′-azino-bis-3(ethylbenzthiazoline-6-sulphonic acid)) and DPPH (2,2-diphenyl-1-picrylhydrazyl) were provided by Sigma Aldrich (Sigma Aldrich, St. Louis, MO, USA). Water was purified on a MilliQ system (Millipore S.A., Molsheim, France).

### 3.2. Gruels Production

Commercial product – dry goji berries were purchased in a local market (imported from China by: Radix-Bis Sp. z o.o., Rotmanka, Poland). Proximate composition of goji berries was as follows in 100 g: caloric value 1384 kJ/330 kcal, fat 3.4 g, carbohydrates 57.9 g, fiber 11.2 g, protein 13.0 g (producer’s data). The tested raw material blends were made from corn grits (DASCA, distributor: Awiko, Lublin, Poland) with an addition of 1, 3 and 5% of dried goji fruits. The blends were moistened up to 14%. The extrudates were prepared by way of a single screw extruder TS-45 (ZMCh Metalchem, Gliwice, Poland) with L/D = 12:1. The range of the temperatures of the extrusion-cooking process was as follows: 125/130/135 °C, respectively in three extruder’s sections. Sample size was 3 kg in total for each experiment with various levels of goji powdered fruits. The extrusion-cooking was carried out at a variable screw speeds: 80, 100 and 120 rpm. A single-open forming die of 3 mm in diameter was used. During the extrusion process, temperature was set in the individual extruder sections at the following profiles: 125–142–132 °C. The obtained extrudates were ground in a laboratory mill LMN10 (TestChem, Radlin, Poland) to obtain gruels with granularity of less than 1 mm.

### 3.3. Preparation of Extracts

The extraction process (sample weight 2 g) was performed in an ultrasonic bath (Bandelin Electronic GmbH & Co. KG, Berlin, Germany) with 40 mL of 80% ethanol for 40 min at a temperature of 60 °C, ultrasound frequency of 33 kHz and a power of 320 W. The extracts were filtered, and 40 mL of 80% ethanol was added to the reminder to repeat the extraction. The obtained extracts were combined, evaporated to dryness and dissolved in 5 mL of methanol [34].

### 3.4. LC-ESI-MS/MS Analysis of Phenolic Acids

Phenolic acids content was determined according to a modified method described by Oniszczuk et al. [35] using an Agilent 1200 Series HPLC system (Agilent Technologies, Santa Clara, CA, USA) connected to a 3200 QTRAP Mass spectrometer (AB Sciex, Redwood City, CA, USA) equipped with electrospray ionization source (ESI). Both were controlled with Analyst 1.5 software (AB Sciex). Separations were carried out on a Zorbax SB-C18 column (2.1 x 100 mm, 1.8 µm particle size; Agilent Technologies) at 20 °C; the injection volume was 3 µL, and the flow rate was 250 µL/min. The gradient method was used with mobile phases: water with 0.1% HCOOH (A) and acetonitrile with 0.1% HCOOH (B), as follows: 0–2 min - 25% B, 3–6 min - 35% B, 8–10 min - 55% B, 12–16 min - 75% B, 19–25 min - 25% B.

ESI operated in the negative-ion mode at the following conditions: capillary temperature 400 °C, curtain gas at 30 psi, nebulizer gas at 50 psi, negative ionization mode source voltage −4500 V. Triplicate injections were made for each standard solution and sample. The analytes were identified by comparing retention time and *m*/*z* values obtained by MS and MS2 with the mass spectra from corresponding standards tested under the same conditions. The identified phenolic acids were quantified based on their peak areas and by comparison with a calibration curve obtained via the corresponding standards.

### 3.5. Antioxidant Properties

#### 3.5.1. Determination of the Total Content of Polyphenolic Compounds (TPC)

The total content of polyphenolic compounds (TPC) was resolved utilizing the modified Folin-Ciocalteu (FC) method [7]. The amount of polyphenols is expressed as mg gallic acid equivalents (GAE) per g of dry mass (d.m.).

#### 3.5.2. Ability to Scavenge DPPH

Measurement of antiradical activity was carried out via DPPH stable radical (2,2-diphenyl-1-picrylhydrazyl) spectroscopy according to the modified method of Burda and Oleszek [7]. Absorbance was measured at 517 nm wavelength, and the UV-VIS spectrophotometer (Genesys UV-VIS, Thermo Scientific, Waltham, MA, US) was calibrated to pure methanol. The measurements were carried out every 5 min for 20 min. This approach allows monitoring changes of absorbance over time and indicates when the plateau is reached. Based on the results, the free radical scavenging ability of the tested extracts was calculated using the following formula:%RSA = [(A_0_ − A_1_)/A_0_] × 100(1)
where: A_0_—the absorbance of the sample except tested extracts; A_1_—the absorbance of the sample with tested extracts.

#### 3.5.3. TLC-DPPH Test

The antioxidant properties of obtained extracts were also examined by applying the TLC-DPPH test. Silica gel plates were used as the stationary phase. A mixture of ethyl acetate, toluene, and formic acid at the ratio of 10:10:0.5 *v*/*v*/*v* was used as a mobile phase. The extracts, along with a reference solution of rutin at the concentration of 0.1 mg/mL, were applied using an automatic TLC applicator Desaga AS-30 (Desaga GmbH, Wiesloch, Germany). The plates were developed in one direction in a horizontal chamber (DS II, Chromdes, Lublin, Poland). After drying, they were sprayed with a 0.1% DPPH methanolic solution. Next, the plates were scanned after 0, 10, and 30 min. The results of TLC-DPPH tests were recorded in the form of JPG documents. For further analysis, we applied the computer software Sorbfil TLC Videodensitometer (Sorbfil, Sorb Polymer Krasnodar, Russia) [36,37]. The TLC-DPPH assay showed the antiradical activity of the analyzed extracts in relation to the activity of 0.1-mg/mL rutin solution (activity of rutin solution equal to “1”).

#### 3.5.4. Trolox Equivalent Antioxidant Capacity (TEAC)

The Trolox equivalent antioxidant capacity (TEAC) assay is based on the ability of the antioxidant to scavenge the blue-green colored ABTS (2,2′-azino-bis(3-ethylbenzothiazoline-6-sulfonic acid) diammonium salt radical cation. This assay is applicable to both lipophilic and hydrophilic antioxidants. The TEAC assay was performed according to Re et al. [38]. The free radical scavenging ability was determined as the µM of Trolox needed to give the same degree of discoloration as the samples (µM TEAC per g of d.m.).

### 3.6. Statistical Analysis

All the measurements were done in three replications; results were mean values of multiple repetitions ± standard deviation (SD). Statistical analysis with ANOVA (Statistica 13.0, StatSoft Inc., Tulsa, OK, USA) was used to determine the significance of differences at α=0.05, with Duncan’s test applied to evaluate the homogenous groups. Pearson’s correlation coefficients and their significance were evaluated at 0.05 and 0.01 for the tested characteristics.

## 4. Conclusions

A new type of functional food product was developed. This is an extruded instant gruel enriched with dry goji fruits. A study of the influence of selected production parameters on the biological properties indicated that increased screw speed allowed obtaining an instant gruel with enhanced antioxidant activity. When processed at high screw speed, increased amounts of goji fruit additive showed a positive effect on free phenolic acids content, especially isoferulic acid. The newly designed dry goji fruit enriched instant gruel can therefore be a valuable source of phenolic compounds accompanied with ready-to-eat convenient features.

## Figures and Tables

**Figure 1 molecules-25-04538-f001:**
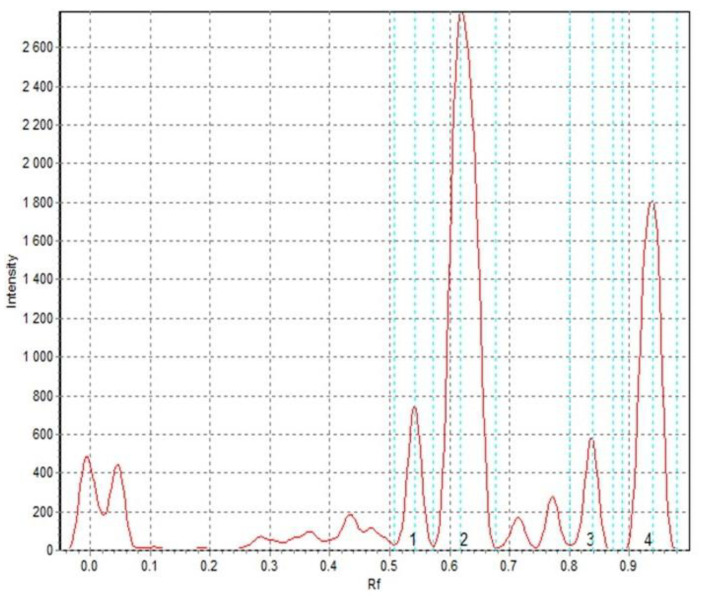
Example TLC chromatogram of compounds with the ability to scavenge DPPH radical (TLC-DPPH test) found in instant gruels with addition of 5% goji fruits; screw speed 100 rpm, plate scanned after 30 min.

**Table 1 molecules-25-04538-t001:** Total content of polyphenolic compounds (TPC), Trolox equivalent antioxidant capacity (TEAC) and results of TLC-DPPH test of instant corn gruels enriched with goji fruits extruded at various screw speeds (n = 3; mean ± SD). The TLC-DPPH assay shows the antiradical activity of analyzed extracts (plate scanned after 30 min) in relation to the activity of 0.1 mg/mL rutin solution (activity of rutin solution equal to “1”).

Addition of Goji Fruits (%)												
Test	80 rpm				100 rpm				120 rpm			
	0%	1%	3%	5%	0%	1%	3%	5%	0%	1%	3%	5%
TPC (mg GAE/mL)	1.13 ^a^ ± 0.07	1.31 ^ab^ ± 0.04	1.63 ^b^ ± 0.08	2.32 ^c^ ± 0.01	1.18 ^a^ ± 0.02	1.39 ^ab^ ± 0.02	1.82 ^bc^ ± 0.08	2.48 ^c^ ± 0.06	1.27 ^ab^ ± 0.03	1.52 ^ab^ ± 0.08	2.03 ^bc^ ± 0.03	2.55 ^c^ ± 0.11
TEAC (µM Trolox/g d.w.)	35.24 ^a^ ± 0.21	36.82 ^ab^ ± 0.71	37.27 ^ab^ ± 1.01	37.89 ^ab^ ± 0.28	36.10 ^a^ ± 0.25	36.94 ^ab^ ± 0.51	37.79 ^ab^ ± 0.32	38.06 ^b^ ± 0.72	35.37 ^a^ ± 0.39	36.22 ^a^ ± 1.24	36.97 ^ab^ ± 0.78	38.48 ^b^ ± 0.99
TLC-DPPH	1.17 ^a^ ± 0.03	1.29 ^ab^ ± 0.02	2.95 ^b^ ± 0.03	3.18 ^b^ ± 0.00	1.28 ^ab^ ± 0.02	1.54 ^ab^ ± 0.03	3.15 ^b^ ± 0.12	3.38 ^bc^ ± 0.07	1.24 ^ab^ ± 0.02	1.65 ^ab^ ± 0.02	3.04 ^b^ ± 0.11	3.51 ^c^ ± 0.08

^a–c^—different letters in rows indicate significant differences at α = 0.05.

**Table 2 molecules-25-04538-t002:** DPPH radical scavenging activity of instant corn gruels enriched with goji fruits depending on time, screw speed and goji fruits addition (n = 3 mean ± SD).

DPPH Radical Scavenging Activity (%RSA)												
Addition of Goji Fruits (%)												
Time (min)	80 rpm				100 rpm				120 rpm			
	0%	1%	3%	5%	0%	1%	3%	5%	0%	1%	3%	5%
10	52.17 ^a^ ± 0.23	60.19 ^ab^ ± 0.43	83.95 ^b^ ± 0.83	91.11 ^bc^ ± 0.42	53.17 ^a^ ± 0.29	87.24 ^b^ ± 0.43	90.95 ^bc^ ± 1.43	94.78 ^c^ ± 0.57	52.74 ^a^ ± 0.12	73.65 ^ab^ ± 0.72	93.95 ^c^ ± 1.43	95.91 ^c^ ± 2.68
15	52.49 ^a^ ± 0.67	61.42 ^ab^ ± 0.36	89.42 ^bc^ ± 1.19	91.11 ^c^ ± 0.87	56.28 ^a^ ± 1.07	87.24 ^bc^ ± 0.72	94.06 ^c^ ± 2.23	96.67 ^c^ ± 0.64	55.67 ^a^ ± 0.29	84.36 ^b^ ± 0.91	94.42 ^c^ ± 1.92	95.91 ^c^ ± 0.64

^a–c^—different letters in rows indicate significant differences at α = 0.05.

**Table 3 molecules-25-04538-t003:** Content of phenolic acids in instant corn gruels enriched with goji fruits (n = 3; mean ± SD); screw speed 100 rpm.

Phenolic Acid		Content of Phenolic Acids (ng/g d.w.)		
	Addition of Goji Fruits (%)			
	0%	1%	3%	5%
protocatechuic	41.4 ^a^ ± 0.31	43.2 ^a^ ± 0.23	61.6 ^b^ ± 0.2	91.6 ^c^ ± 0.4
*trans*-caffeic	ND^†^	BQL ^††^	BQL ^††^	46.4 ^a^ ± 0.1
4-OH-benzoic	305.6 ^a^ ± 3.4	428.1 ^ab^ ± 2.5	468.0 ^b^ ± 0.7	664.3 ^c^ ±3.2
gentisic	ND ^†^	BQL ^††^	BQL ^††^	18.2 ^a^ ± 0.0
*p*-coumaric	ND ^†^	412.3 ^a^ ± 1.5	712.1 ^b^ ± 1.1	1644.1 ^c^ ± 3.5
ferulic	143.6 ^a^ ± 1.2	172.4 ^ab^ ± 0.4	282.2 ^b^ ± 1.1	684.2 ^c^ ± 2.4
isoferulic	7780.6 ^a^ ± 6.7	8720.1 ^ab^ ± 4.8	8880.2 ^ab^ ± 2.9	9120.1 ^b^ ± 3.1
salicylic	214.0 ^a^ ± 2.9	240.0 ^ab^ ± 1.2	250.4 ^b^ ± 0.3	508.4 ^c^ ± 2.2
**sum**	**8485.2**	**10016.1**	**10654.5**	**12777.3**

^a–c^—different letters in columns indicate significant differences at α = 0.05; ND ^†^—not detected; BQL ^††^—below quantification level.

**Table 4 molecules-25-04538-t004:** Analytical parameters of LC-MS/MS quantitative method; data for calibration curves, limit of detection (LOD) and limit of quantification (LOQ) values for each analyzed phenolic acid.

Phenolic Acid	Regression Equation	LOD [ng/mL]	LOQ [ng/mL]	*r^2^*	Linearity Range [ng/mL]
protocatechuic	y = 86.3x + 1240	10	25	0.9999	50–12500
*trans*-caffeic	y = 1080x + 6640	10	25	0.9992	25–2500
4-OH-benzoic	y = 1470x + 7020	20	50	0.9999	50–2000
gentisic	y = 717x + 45100	5	15	0.9994	15–5000
*p*-coumaric	y = 888x + 855	10	25	0.9999	25–2500
ferulic	y = 360x − 3740	10	25	0.9994	40–2000
isoferulic	y = 360x − 3740	10	25	0.9994	40–2000
salicylic	y = 2060x + 13000	10	30	0.9999	30–1000

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
