# Peer review of "Polyphenol Composition and Antioxidant Potential of Instant Gruels Enriched with Lycium barbarum L. Fruit"

_molecules, 2020, doi:10.3390/molecules25194538_

Round 1

Reviewer 1 Report

This study describes the effect of Goji berries addition together with processing conditions on the polyphenol content and antioxidant properties of gruels.

The manuscript would need a deep revision before being considered for publication.

First of all, the full manuscript should be revised by a native english speaker.

The manuscrpt is not well structured, with material and methods issues included and/or repeated in the results section (i.e. lines 111-113; lines 124-130…). Sometimes the discussion of the results comes before their description and makes the reading difficult to follow (i.e. lines 90-110…). In addition, the discussion of the results should be improved and well focused.

Some issues should be further explained: why do the authors only quantify phenolic acids? Why do they choose these specific phenolic acids? There is no discussion concerning the polyphenolic profile of goji berries described in literature…

If the authors speak about a correlation between two variables, they should indicate r and p value…

If it is a combined results and discussion section, it sould be indicated.

Table 2 could be reduced to final point measurement.

Lines 59-62: some references supporting these sentences should be included.

How do the authors conclude the effect of processing? In table 1 (foot note) it is said that different letters indicate significant differences in a row. Like this, there is no difference in the TPC between speeds (for each % of added berries…). For example, 5% berries in 80rpm, 100rpm and 120rpm have a “c”, so they are no significantly different, are they?… there is no effect of speed then, is there?

Keywords. Some keywords are repeated, since LC = liquid chromatography. Some of they are not specific of the manuscriot: “gluten-free gruels”? there is no reference to this specific question along the manuscript…

Quality of figure 2 should be improved.

Author Response

Response for Reviewer 1

The authors would like to thank the Reviewer for the valuable comments which have helped to improve the quality of the manuscript.

This study describes the effect of Goji berries addition together with processing conditions on the polyphenol content and antioxidant properties of gruels. The manuscript would need a deep revision before being considered for publication. First of all, the full manuscript should be revised by a native english speaker.

Revision by a native speaker (Jack Stanley Dunster from Canada, Language Editor of Current Issues in Pharmacy and Medical Sciences) was done before re-submission.

The manuscrpt is not well structured, with material and methods issues included and/or repeated in the results section (i.e. lines 111-113; lines 124-130…). Sometimes the discussion of the results comes before their description and makes the reading difficult to follow (i.e. lines 90-110…). In addition, the discussion of the results should be improved and well focused.

Section Results and Discussion has been improved. Methods issues have been removed from this section. The discussion was completed (indicated in the manuscript) and now it is preceded by a description of the results.

Some issues should be further explained: why do the authors only quantify phenolic acids? Why do they choose these specific phenolic acids? There is no discussion concerning the polyphenolic profile of goji berries described in literature…

Many health beneficial effects (e.g. antibacterial, chemopreventive, antiviral, anti-inflammatory and antioxidant) have been acknowledged for plant-derived food products rich in phenolic acids (both benzoic and cinnamic acid derivatives) [Sova & Saso, 2020 (https://doi.org/10.3390/nu12082190); Olech et al. 2020 (doi:10.3390/molecules25081804)].

Phenolic acids (PA) were found to be a significant fraction of the polyphenols present in L. barbarum L fruit, constituting 32.7 mg/g of their dry weight [Mocan et al. 2019 (https://doi.org/10.3390/antiox8110562)]. Goji fruits were previously reported to contain biologically active PA, including benzoic and cinnamic acid derivatives, e.g. hydroxybenzoic acids, caffeic and p-coumaric acid.

The amount of PA may be affected by processing of raw material. Therefore, content of PA is commonly determined during investigation and quality control of new functional foods or plant-derived food components. That is why we aimed to determine the influence of the screw speed on qualitative and quantitative polyphenols profile and antioxidant properties of newly created extruded gruels. For this reason, the applied LC-MS method was optimized to contain  the aforementioned goji metabolites.

This information was added to Introduction.

If the authors speak about a correlation between two variables, they should indicate r and p value…

Proper values of r and p have been added to show correlations between the tested properties.

If it is a combined results and discussion section, it sould be indicated.

Section 2 is now referred to as Results and Discussion as it not only reports the results obtained, but rather compares and discusses them with those reported in the literature.

Table 2 could be reduced to final point measurement.

Table 2 has been reduced to final point measurement.

Lines 59-62: some references supporting these sentences should be included.

References have been included.

How do the authors conclude the effect of processing? In table 1 (foot note) it is said that different letters indicate significant differences in a row. Like this, there is no difference in the TPC between speeds (for each % of added berries…). For example, 5% berries in 80rpm, 100rpm and 120rpm have a “c”, so they are no significantly different, are they?… there is no effect of speed then, is there?

We observed only slight effect of screw speed this time, especially on antioxidant activity. This might be because we used similar low raw material moisture content and high treatment temperature to achieve almost completely gelatinized instant gruels. Proper information was added in the main text.

  1. 95-96: However, increasing the screw speed during instant gruel plus goji processing has no significant effect on the content of TPC.
  2. 107-110: The change of screw speed applied during the instant gruel extrusion-cooking (indicated by different letters in Table 1) was shown to slightly affect the intensity of antioxidant activity of the tested instant gruels – especially during the first ten minutes.
  3. 114-115: Increasing screw speed had only a slight effect on TEAC activity, especially with high content of additive.

Keywords. Some keywords are repeated, since LC = liquid chromatography. Some of they are not specific of the manuscriot: “gluten-free gruels”? there is no reference to this specific question along the manuscript…

Keywords have been changed. Phrases „LC-ESI-MS/MS” and „gluten-free gruels” have been removed.

Quality of figure 2 should be improved.

Quality of figure 2 has been improved and inserted as Figure S1 in Supplementary material (as recommended by the Reviewer 2).

Reviewer 2 Report

The authors describe a process for obtaining an instant gruel enriched with dry goji berries as a functional food. In addition, they assess the influence of extruder screws speed and process temperature on the antioxidant composition of the blends.

The work is interesting. The experimental data obtained are congruent with the methods described in the Materials and methods section, and the Conclusion section clear presents experimental results obtained, highlighting trends and points of interest. However, there are some points that need some minor tweaks.

- Section 2 (Results) should be referred to as Results and Discussion as it not only reports the results obtained, but rather compares and discusses them with those reported in the literature.

- The progression of the number of citations in the manuscript is not congruent with the References section. On line 91 the numbering of the citations is inconsistent with the previous citations (line 75). Maybe in a first draft the authors moved the Material and Methods section to the end of the manuscript, without consequently re-numbering the citations.

- If the managing Editor agrees I suggest representing Tables 1 and 2 in landscape mode for a better reading of the reported data.

- The chromatograms of Fig. 2 are of poor quality. If authors do not have any better I suggest to insert such Figure as Supplementary material as they are not essential for understanding the results.

- Please, check References n. 4, 7 for italics species names, and 23 for underlined authors name.

Author Response

Response for Reviewer 2

The authors would like to thank the Reviewer for the valuable comments which have helped to improve the quality of the manuscript.

The authors describe a process for obtaining an instant gruel enriched with dry goji berries as a functional food. In addition, they assess the influence of extruder screws speed and process temperature on the antioxidant composition of the blends.

The work is interesting. The experimental data obtained are congruent with the methods described in the Materials and methods section, and the Conclusion section clear presents experimental results obtained, highlighting trends and points of interest. However, there are some points that need some minor tweaks.

- Section 2 (Results) should be referred to as Results and Discussion as it not only reports the results obtained, but rather compares and discusses them with those reported in the literature.

Section 2 is now referred to as Results and Discussion

- The progression of the number of citations in the manuscript is not congruent with the References section. On line 91 the numbering of the citations is inconsistent with the previous citations (line 75). Maybe in a first draft the authors moved the Material and Methods section to the end of the manuscript, without consequently re-numbering the citations.

The citation numbers in the manuscript and References section have been corrected.

- If the managing Editor agrees I suggest representing Tables 1 and 2 in landscape mode for a better reading of the reported data.

Tables 1 and 2 are represented in landscape mode.

- The chromatograms of Fig. 2 are of poor quality. If authors do not have any better I suggest to insert such Figure as Supplementary material as they are not essential for understanding the results.

The quality of Figure 2 has been improved. Figure 2 has been added as Supplementary Material (Figure S1).

- Please, check References n. 4, 7 for italics species names, and 23 for underlined authors name.

References have been corrected.

Reviewer 3 Report

No any special comment or suggestion. The paper is very correct, consisting of all necessary elements, providing a new principle in food processing.

Author Response

Reviewer 3

No any special comment or suggestion. The paper is very correct, consisting of all necessary elements, providing a new principle in food processing.

We would like to thank the Reviewer for his/her favourable opinion.

Reviewer 4 Report

The manuscript authored by Olech et al entitled "Impact of production parameters on polyphenols 2 composition and  antioxidant potential of instant gruels enriched with Lycium barbarum L fruit” is clear, well written, precise and easy to follow.

In this manuscript, authors investigated the production of a kind of porridge from corn with some portions of dried goji and determined total phenolic content and its antioxidant proprieties (from extruded samples). In addition, authors tried to identify the optimum rotation and other parameters for the extrusion process.

Gruel from goji fruits (purchased from a local market) was prepared by using ultrasound and extraction with EtOH 80%. Analysis of Phenolic Acids were performed by LC-ESI-MS/MS (3200 QTRAP Mass spectrometer (AB Sciex, Redwood City, CA, USA) equipped with 275 electrospray ionization source (ESI negative mode). Antioxidant tests included: (i) Determination of the Total Content of Polyphenolic Compounds (TPC); (ii) Ability to Scavenge DPPH; (iii) TLC-DPPH Test and (iv) Trolox Equivalent Antioxidant Capacity (TEAC).

Eight phenolic acids were present in samples, derived from benzoic acid and include: protocatechuic acid, 4-OH-benzoic acid, gentisic acid, salicylic acid, and cinnamic acid derivatives: rans-caffeic, p-coumaric, ferulic, and isoferulic. The more concentration of phenolics, the more efficient to scavenge reactive oxygen species.

Quite robust experiments were involved in these studies; however, the work is solid. References were correctly selected and cited along the text.

Major/Minor Concerns: (i) The phenolic determination considered samples only from local market, I wonder if there are more variation with samples from other places and region. Authors should consider that. It is not totally clear this part in the manuscript. (ii) Authors need to clarify the extrusion experiments and controls used.

The figures and table are well-distributed, presented in good graphical quality and include the main results authors found in this manuscript. This manuscript needs to be carefully revised by either a native English speaker or a professional language editing service to improve the grammar and readability.

To finish this revision, besides these above mentioned suggestions, the manuscript is good. This work is an important contribution to our knowledge of phenolic compounds and enrichment in food process.

Author Response

Reviewer 4

The authors would like to thank the Reviewer for the valuable comments which have helped to improve the quality of the manuscript.

The manuscript authored by Olech et al entitled "Impact of production parameters on polyphenols 2 composition and  antioxidant potential of instant gruels enriched with Lycium barbarum L fruit” is clear, well written, precise and easy to follow.

In this manuscript, authors investigated the production of a kind of porridge from corn with some portions of dried goji and determined total phenolic content and its antioxidant proprieties (from extruded samples). In addition, authors tried to identify the optimum rotation and other parameters for the extrusion process.

Gruel from goji fruits (purchased from a local market) was prepared by using ultrasound and extraction with EtOH 80%. Analysis of Phenolic Acids were performed by LC-ESI-MS/MS (3200 QTRAP Mass spectrometer (AB Sciex, Redwood City, CA, USA) equipped with 275 electrospray ionization source (ESI negative mode). Antioxidant tests included: (i) Determination of the Total Content of Polyphenolic Compounds (TPC); (ii) Ability to Scavenge DPPH; (iii) TLC-DPPH Test and (iv) Trolox Equivalent Antioxidant Capacity (TEAC).

Eight phenolic acids were present in samples, derived from benzoic acid and include: protocatechuic acid, 4-OH-benzoic acid, gentisic acid, salicylic acid, and cinnamic acid derivatives: rans-caffeic, p-coumaric, ferulic, and isoferulic. The more concentration of phenolics, the more efficient to scavenge reactive oxygen species.

Quite robust experiments were involved in these studies; however, the work is solid. References were correctly selected and cited along the text.

Major/Minor Concerns: (i) The phenolic determination considered samples only from local market, I wonder if there are more variation with samples from other places and region. Authors should consider that. It is not totally clear this part in the manuscript. (ii) Authors need to clarify the extrusion experiments and controls used.

Certainly, there would be differences between raw plant materials from different places and regions (https://doi.org/10.3390/antiox8110562). Almost all Goji berries on the Polish commercial market are imported. Since production of instant gruels takes place under industrial conditions, we decided to analyze fruits available in industrial quantities. Description of raw materials has been improved: dry goji berries were purchased in a local market (imported from China by: Radix-Bis Sp. z o.o., Rotmanka, Poland). Proximate composition of goji berries was as follows in 100 g: caloric value 1384 kJ/330 kcal, fat 3.4 g, carbohydrates 57.9 g, fiber 11.2 g, protein 13.0 g (producer’s data). We added proper description into manuscript.

The extrusion experiment is described but we added some details about the extrusion-cooking process conditions: The range of the temperatures of the extrusion-cooking process was as follows: 125/130/135°C, respectively in three extruder’s sections. Sample size was 3 kg in total for each experiment with various levels of goji powdered fruits. We do hope that additional description is sufficient.

The figures and table are well-distributed, presented in good graphical quality and include the main results authors found in this manuscript. This manuscript needs to be carefully revised by either a native English speaker or a professional language editing service to improve the grammar and readability.

To finish this revision, besides these above mentioned suggestions, the manuscript is good. This work is an important contribution to our knowledge of phenolic compounds and enrichment in food process.

Reviewer 5 Report

Comments for molecules-900265

The manuscript focuses on production of innovative instant corn gruels with various contents of dry goji berries, and determination of the content of polyphenolic compounds, including individual free phenolic acids, as well as the antioxidant properties of analyzed extracts, and identification of the optimum of one of the most important production parameters- the rotational speed of the extruder’s screw during gruel processing. The content of the paper is quite valuable, but there are some problem in experimental design should be solved before the manuscript been considered for publication.

Substantial revisions

Q1: In Table3, 0~3% trans-caffeic shows no detection or lower than the level. Why is there a significant difference in asterisk?

Q2:  The research topic is antioxidant potential of instant gruels enriched with Lycium barbarum L fruit. However, In addition to DPPH assay, there are more than one anti-oxidation ability to detect phenolic content, such as ORAC, BHT, SOD.. etc. What is the reason that only DPPH assay was used for the experiment?

Q3: Try to discuss and compare the ability in the antioxidant system represented by the same ingredient extract in other literatures using different antioxidant test methods.

Author Response

Reviewer 5

The authors would like to thank the Reviewer for the valuable comments which have helped to improve the quality of the manuscript.

The manuscript focuses on production of innovative instant corn gruels with various contents of dry goji berries, and determination of the content of polyphenolic compounds, including individual free phenolic acids, as well as the antioxidant properties of analyzed extracts, and identification of the optimum of one of the most important production parameters- the rotational speed of the extruder’s screw during gruel processing. The content of the paper is quite valuable, but there are some problem in experimental design should be solved before the manuscript been considered for publication.

Substantial revisions

Q1: In Table 3, 0~3% trans-caffeic shows no detection or lower than the level. Why is there a significant difference in asterisk?

We are very sorry for this mistake, asterisk is usually used to indicate significant differences, we used that symbol to explain descriptions below the table. Now in Table 3 we changed symbols to indicate “not detected” and “below quantification level” into following: ND†  – not detected; BQL††– below quantification level

Q2:  The research topic is antioxidant potential of instant gruels enriched with Lycium barbarum Lfruit. However, In addition to DPPH assay, there are more than one anti-oxidation ability to detect phenolic content, such as ORAC, BHT, SOD.. etc. What is the reason that only DPPH assay was used for the experiment?

At this stage of research, we wanted to observe the direction and degree of bioactivity changes depending on the preparation method. We decided to use three antioxidant assays: ability to scavenge DPPH radical, TLC-DPPH and evaluation of Trolox Equivalent Antioxidant Capacity with ABTS radical. These methods are good tools for the rapid monitoring of sample’s activity. Spectrophotometric assay with DPPH radical is more suitable for analysis of polar compounds, while ABTS radical enables evaluation of activity both lipophilic and hydrophilic ingredients. TLC-DPPH, in turn, is a combination of chromatography and bioactivity analysis, which allows to initially indicate compounds (or groups of compounds) with the highest antioxidant activity (https://doi.org/10.1016/S0891-5849(98)00315-3; https://doi.org/10.1016/j.foodchem.2011.10.067). All three assays provided complementary information how production method influences gruels’ antioxidant potential.

Q3: Try to discuss and compare the ability in the antioxidant system represented by the same ingredient extract in other literatures using different antioxidant test methods.

Dried and freeze-dried goji fruits were previously reported to contain ferric reducing antioxidant agents with FRAP values 0.15 - 0.62 and  23.09 mmol Fe2+/kg, respectively (Xin et al. 2017; Donno et al. 2019). High antioxidant capacity of L. barbarum berries was also demonstrated by the CUPRAC (cupric ion reducing antioxidant capacity) (26.91 - 35.41 mg TE/g), oxygen radical absorbance capacity (ORAC) assay (180-260 μmol vitamin C equivalents/ g) and electrochemical methods (Protti et al. 2017; Mocan et al. 2018).

This information was added to the Results section.

NEW REFERENCES

Donno, D.; Mellano, M.G.; Riondato, I.; De Biaggi, M.; Andriamaniraka, H.; Gamba, G.; Beccaro, G.L. Traditional and unconventional dried fruit snacks as a source of health-promoting compounds. Antioxidants 2019, 8, 396. https://doi.org/10.3390/antiox8090396

Mocan, A.; Moldovan, C.; Zengin, G.; Bender, O.; Locatelli, M.; Simirgiotis, M., Atalay, A.; Vodnar, D.C.; Rohn, S.; Crișan, G. UHPLC-QTOF-MS analysis of bioactive constituents from two Romanian goji (Lycium barbarum L.) berries cultivars and their antioxidant, enzyme inhibitory, and real-time cytotoxicological evaluation. Food Chem. Toxicol. 2018, 115, 414-424. doi:10.1016/j.fct.2018.01.054

Protti, M.; Gualandi, I.; Mandrioli, R.; Zappoli, S.; Tonelli, D.; Mercolini, L. Analytical profiling of selected antioxidants and total antioxidant capacity of goji (Lycium spp.) berries. J. Pharm. Biomed. Anal. 2017, 143, 252-260. doi:10.1016/j.jpba.2017.05.048

Xin, G.; Zhu, F.; Du, B.; Xu, B. Antioxidants distribution in pulp and seeds of black and red goji berries as affected by boiling processing. J. Food Qual. 2017, 3145946.  doi:10.1155/2017/3145946

Round 2

Reviewer 5 Report

Comments for molecules-900265

The manuscript focus on production of innovative instant corn gruels with various contents of dry goji berries, and determination of the content of polyphenolic compounds, including individual free phenolic acids, as well as the antioxidant properties of analyzed extracts, and identification of the optimum of one of the most important production parameters- the rotational speed of the extruder’s screw during gruel processing. The content of the paper is quite valuable. In the revised manuscript, My questions in experimental design has been responded by authors. Thus I suggest that the manuscript be considered for publication.